# Morphological and Surface Microtopographic Features of HPHT-Grown Diamond Crystals with Contact Twinning

Kaiyue Sun [1], Taijin Lu [2,*], Mingyue He [1,*], Zhonghua Song [2], Jian Zhang [2] and Jie Ke [2]

1 School of Gemmology, China University of Geosciences, Beijing 100083, China
2 National Gems & Jewelry Testing Co., Ltd., Beijing 100013, China
* Correspondence: taijinlu@hotmail.com (T.L.); hemy@cugb.edu.cn (M.H.); Tel.: +86-18801024609 (T.L.); +86-13801191820 (M.H.)

**Abstract:** Gem-grade twinned high-pressure high-temperature (HPHT) synthetic diamond crystals are rare. Hence, few investigations on their morphological features and formation have been reported. In this article, the morphological and surface microtopographic features of HPHT synthetic-diamond crystals contact twinning is detailed and investigated. It indicates that twins of diamond forming and nucleating during the early stages of the growth and the development of {100} and {111} growth sectors on either side of such boundaries proceeds independently, which affects the final morphology of the diamond crystals. According to the different features of crystal macroscopic morphological properties, two kinds of twin model have been established. The formation of twin crystals changed the lattice of diamonds with face-centered cubic dimensions. The type of diamond lattice at the twin boundary is hexagonal and closely packed, which has potential for further developing the application of synthetic diamond twin crystals.

**Keywords:** crystal morphology; surface microtopographic features; contact twinning; HPHT diamond



## 1. Introduction

The morphology of diamonds has always been a hot topic, including the shape, size, and the number of crystals, which can reflect the growth conditions and provide information about the formation process. The classification and origin of diamond morphology have been studied and discussed for many years. Among them, Orlov (1977) classified diamonds into 10 types according to the different formation process. I~V are monocrystalline diamonds, including octahedral diamonds, diamond cubes, diamond cubes with a large number of microscopic inclusions surrounding them, coated diamonds, and octahedral crystals with a large number of syngenetic graphite inclusions. VI~X are polycrystalline diamonds, including spherical diamonds often with a radial fringe structure, translucent diamonds with parallel intergrowths, with clear grain boundary aggregates, and without clear grain boundary aggregates. Cryptocrystallines consist of submicroscopic diamond particles [1]. Other, more generalized diamond classification schemes emphasize the growth mechanisms that produce the different morphological types of diamonds. Based on the differences in morphology and growth mechanisms, R Tappert (2010) distinguished three principle types of diamond: monocrystalline diamonds, fibrous, and polycrystalline [2].

Previous studies conducted on the morphology of diamonds reveal the intricacy of the growth mechanisms. Japanese scholar Ichiro Sunagawa (2004) summarized the relationship between the growth speed and driving force of diamonds. Under the different driving forces, these correspond to three growth mechanisms, respectively, including the block-adhesive growth mechanism under high driving forces, two-dimensional nucleation growth mechanism under medium driving forces, and dislocation (spiral) growth mechanism under low driving forces. From high to the low driving forces, the morphology of a diamond crystal changes from spherulite, dendritic, and skeleton crystals to a polyhedron [3]. Natural

diamonds occur as single crystals, as aggregates of a few crystals, or as clusters of countless, small crystallites. The external shape, or habit, of most naturally occurring monocrystalline diamonds is restricted to three basic geometric forms: the octahedron, the cube, and the rhombic dodecahedron. However, there are many special circumstances that can change the morphology of diamonds and produce aggregates, parallel intergrowths, and twins.

Twinning may occur by growth, thermal transformation, or mechanical deformation. Diamond twins only occur during growth and only by reticular merohedry [4,5]. In terms of morphology, the twins in diamonds mainly include contact twinning and an interpenetrant twin. They consist of two or more intergrown diamond crystals. Additionally, the individual crystals of twins are oriented in different directions [6]. A. Yacoot (1998) proposed three patterns of natural diamond growth, (i) faceted growth on {111} planes; (ii) cuboid non-faceted growth on approximate {100} planes; and (iii) fibrous growth along <111> directions, and deduced that when twinning occurs, the mode (i) produces contact twinning and (ii) an interpenetrant twin [5,7]. The individual crystals in twin diamonds can be connected along external crystal faces, in which case they present contact twinning. When they are connected along internal planes, they are referred to as interpenetrant twins [8]. Lu Taijin et al. (2018) made some observations on the surface dissolution features of natural-diamond contact twinning and observed hexagonal pits; rhombic pits along with greater symmetry were observed at the twin boundaries together with dislocations [9]. W. G. Machado (1998) considered that the interpenetrant twin in a natural diamond is usually colored and appears to have been formed by fibrous growth [8,10].

Twinning crystals has a great influence on the properties of diamonds. However, the formation of twinning in natural diamonds is relatively minor. Therefore, many researchers attempted to control the formation of twinning by the synthetic method. At present, the common synthetic diamond methods are high temperature and high pressure (HPHT) and chemical vapor deposition (CVD). Most previous studies conducted on synthetic diamond twin crystals address the synthesis of nano-sized polycrystalline diamond by the HPHT method and the microregion twinning phase of diamond films synthesized by CVD.

The following results were achieved. The hardness of diamonds can be enhanced by nano-structuring by means of nanograined and nano-twinned microstructures. For example, a nanograined diamond (ng diamond) with grain sizes of 10–30 nm has been reported as high as 110–140 GPa in Knoop hardness, significantly higher than that of a single crystal diamond. A nanotwinned diamond (nt diamond) with an average twin thickness ($\lambda$) of 5–8 nm, synthesized by compressing onion-structured precursors, was recently reported to possess a Vickers hardness of 175–200 GPa, setting a new world record [11]. The crystal twins in CVD diamonds originate from the formation of hydrogen-terminated four-carbon atom clusters on a local {111} surface morphology, the result of the hydrogen defects on the octahedral plane [12–16].

Since gem-quality twinned synthetic diamond crystals are rare, there have been few investigations conducted on them. In order to comprehend the formation of twinning in diamonds, an enhanced understanding and guidance of the synthesis and application are required. In this study, a set of HPHT synthetically grown diamond crystals produced by a Chinese company with contact twinning are observed with the morphological and surface microtopographic features for studying the growth mechanism.

## 2. Materials and Methods

A total of 18 diamond crystals with contact twinning synthesized by high temperature and high pressure were provided by Zhengzhou Sino-Crystal Diamond Co., Ltd., Zhengzhou, China. They were labeled as TW-01~TW-18, respectively (Figure 1). The synthetic pressure and temperature were 5.4 $\pm$ 0.2 Gpa and 1350 $\pm$ 10 °C, respectively. The color of the samples was brownish yellow due to the presence of nitrogen in the growth process. High-purity titanium metal was used to obtain nitrogen during the diamond-crystallization process. High-purity graphite powder (99.99%) and an iron-nickel alloy were used as the carbon source and solvent, respectively. Among the samples,

the largest one, TW-01, weighed 1.021 ct, while the others, TW-02~TW-18, weighed between 0.1655 ct–0.231 ct. The crystal morphology was different from the traditional HPHT synthetic-diamond products, which are cube-octahedral structures. All of the samples presented a symmetrical crystal morphology. It could be observed that they all had a re-entrant angle, which is a feature of twinning, and that the orientation of the same crystal plane on both sides of the twin boundary changed; therefore, the samples were very rare. In order to discuss their morphological and surface microtopographic features and further explore their causes, varieties of observation techniques were used to produce the analysis of the morphology features of HPHT-grown synthetic diamonds with contact twinning.

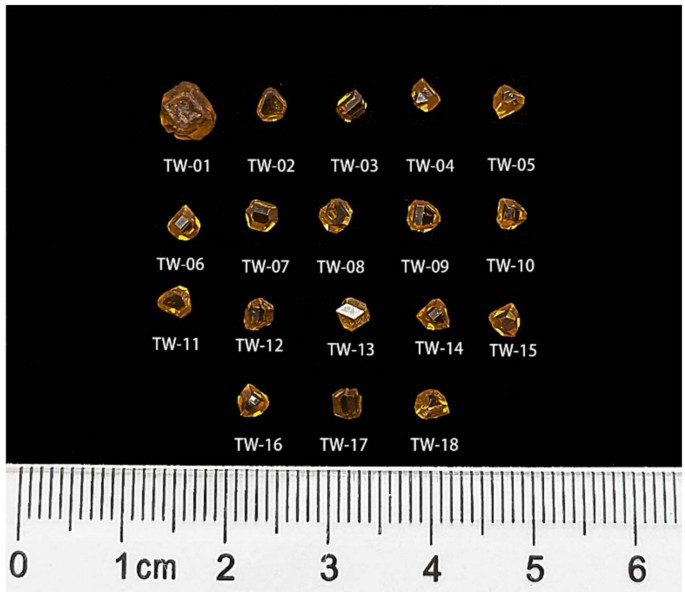

**Figure 1.** Investigated samples.

The crystal morphology, surface microtopographic features, including etch figures and growth features of all the samples were examined by gemological microscopy with a CCD camera system and 3D Laser Confocal Microscopy (OLS5000-SAF) with a color imaging optical system and laser confocal optical system. We selected typical features for discussion to reveal their formation information. The samples, including TW-01, TW-04, TW-07, and TW-12, were examined further at greater magnifications with a Zeiss Supra 55 field emission scanning electron microscope (FESEM) under 20 kV. Samples were coated with 5 nm thick carbon. Sample TW-09 was measured by X-ray Fluorescent Spectroscopy (micro-XRF) to observe the distribution of elements. The system worked at 50 kV and 600 µA and the spot size was <20 µm.

## 3. Results

### 3.1. Macroscopic Morphology

Unlike the cubic octahedral single crystal of traditional HPHT-grown synthetic diamonds, the crystal morphology of the twinning crystals was more complex. In the majority of samples, each single crystal of contact twinning resembled the shape of a tower. Additionally, the macroscopic morphology showed a symmetrical crystal with a morphology that reflected the mirror symmetry of the twin boundary or of a less symmetrical crystal arising from differences in the number and development of the {100} and {111} growth sectors on either side of the twin boundary. The (111) plane on either side of the twin boundary formed the re-entrant angle (Figure 2a,b) [9,17]. There were two samples, TW-02 and TW-04, similar to the octahedral Macle of natural diamonds (Figure 2c,d) [18]. Their morphology has a good mirror symmetry on both sides of the twin boundary. Moreover, two special samples, TW-07 and TW-12, had slightly different features with others at the

twin boundary where the single crystal of the twinning made contact with the planes of (100) and (111). Additionally, the twin boundary of the TW-07 sample seemed to be a certain width showing that it did not have a macroscopic two-dimensional boundary, but rather a transition region (Figure 2e). Moreover, the twin boundary of the TW-13 sample was not always continuous and sometimes blocked by other small facets (Figure 2f). Among the samples, only the TW-01 sample had a seed on the (100) plane (Figure 2g). The morphology features around the seed were rough. The crystal plane in the TW-07 sample was not always smooth and often developed crystal steps (Figure 2h).

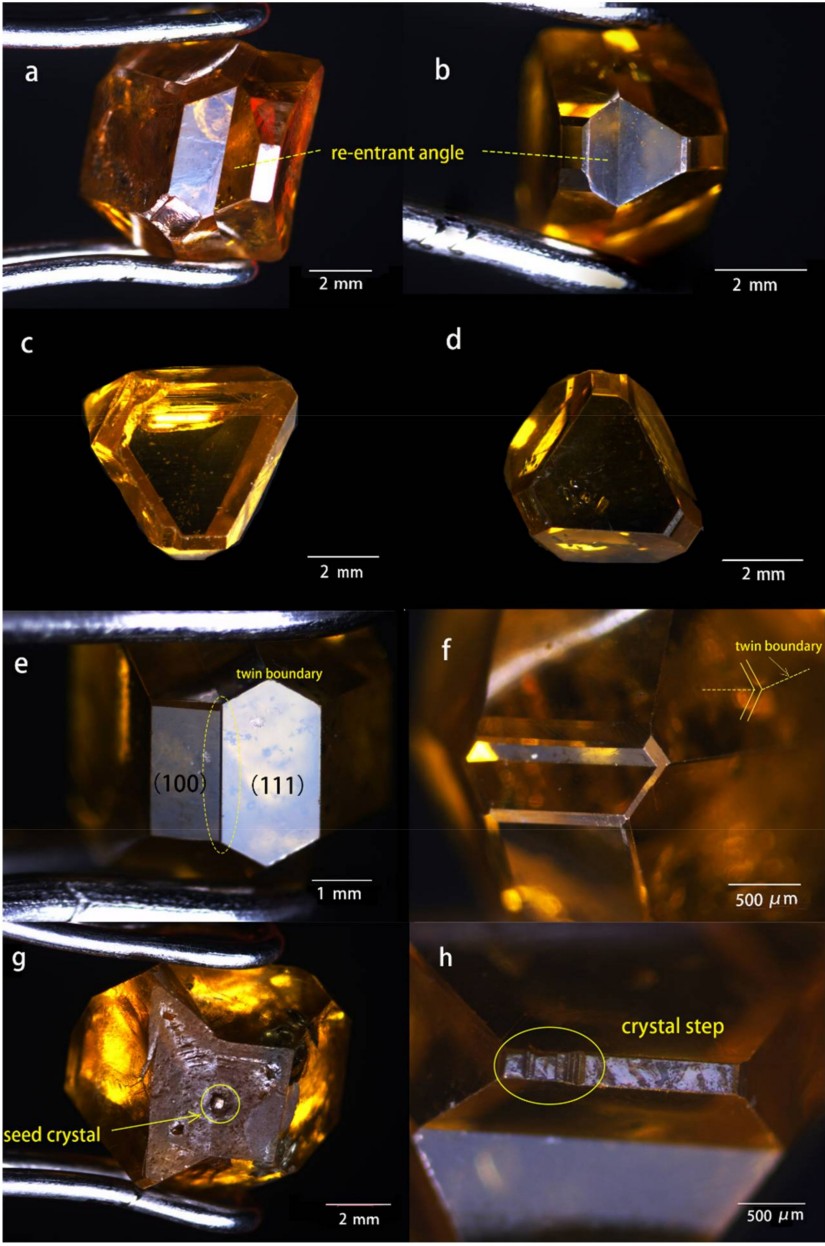

**Figure 2.** Macroscopic morphology of HPHT synthetic diamond with contact twinning: (**a**,**b**) the features of the re-entrant angle in TW-01 and TW-02; (**c**,**d**) the crystal morphology of TW-02 and TW-04, which are similar to the octahedral Macle of a natural diamond; (**e**,**f**) the features of twin boundaries in TW-07 and TW-13; (**g**) seed crystal of TW-01; (**h**) the crystal steps in TW-07.

Observing the macroscopic morphology of synthetic-diamond contact twinning TW-01, TW-04, TW-07, and TW-12 by the SEM, the contact relationship of the crystal plane was expressed more clearly. As shown in Figure 3a, crystal domains share a common

{111} plane, which are offset from one another to create surface faceting with re-entrant angles. Arising from the differences in the number and development of the {100} and {111} growth sectors on either side of the twin boundary, the degree of crystal symmetry is varying. Figure 3b shows that the sample has features of crystal morphology similar to octahedral Macle in natural diamonds. Macle crystal is the fattened octahedral crystals that are connected along an octahedral plane, with one of the crystals being oriented in the opposite direction. This orientation can be viewed as one crystal being rotated in the contact plane by 180° with respect to the other crystal. The morphology bounded only by (111) faces which grow by the layer-by-layer [18,19]. Finally, due to the resorption, Macle crystals can convert into flattened dodecahedral diamonds [2,20]. The difference is that HPHT synthetically grown diamond crystals with contact twinning do not undergo the flattening process. The morphology of twinning is clear to distinguish. As shown in Figure 3c,d, the single crystal of twining makes contacts with the planes of (100) and (111). Additionally, the twin boundary seems to be a certain width. The single crystal still has the features of a tower shape.

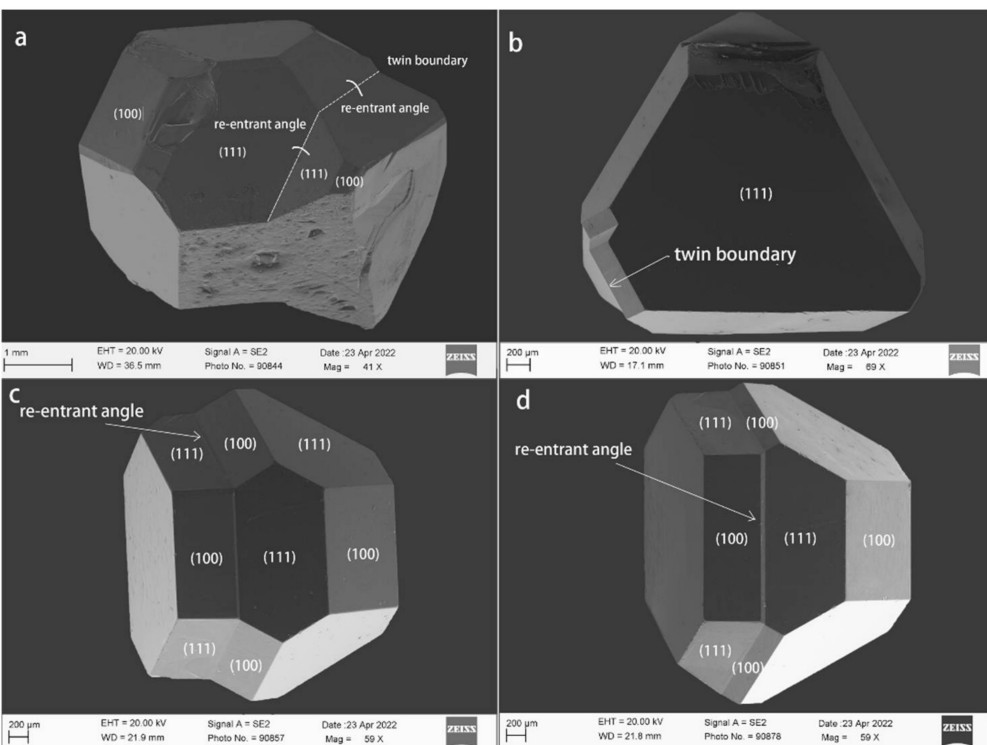

**Figure 3.** The macroscopic morphology of HPHT synthetically grown diamond crystals with contact twinning observed by SEM: (**a**) the features of the symmetrical crystal morphology and re-entrant angle in TW-01; (**b**) the sample of TW-04 has features of crystal morphology similar to octahedral Macle in natural diamonds; (**c**,**d**) the single crystal of twining makes contacts with the planes of (100) and (111) in TW-07, TW-12.

*3.2. Twin Boundary*

The formation of a twin boundary is often accompanied by the appearance of re-entrant angles. In the current study, three basic appearances of these twin boundaries were observed: (i) The contact twinning domains shared a common {111} plane. The intersecting lines of two single crystal (111) planes formed the twin boundary (Figure 3a). (ii) The single crystal of twining makes contact with the planes of (100) and (111), and it seemed to be a certain width of the twin boundary (Figure 3c,d). (iii) The twin boundary was not always continuous and was sometimes blocked by other small facets (Figure 2f). Grain boundaries can be divided into coherent and incoherent factors. The only coherent boundary between the crystal and twin is the (111) plane. Consequently, the twin boundary

of (111) contact twinning in diamonds is coherent, which can be observed as linear features by gemological microscopy [21]. When the twin boundary was observed by SEM, it was not a sharp, straight line similar to the crystal edge, but is similar to the shape of the pipe with a different height to the crystal plane (Figure 4a–e). The twin boundary forms a zigzag structure when the twining contacts the planes of (100) and (111) (Figure 4f).

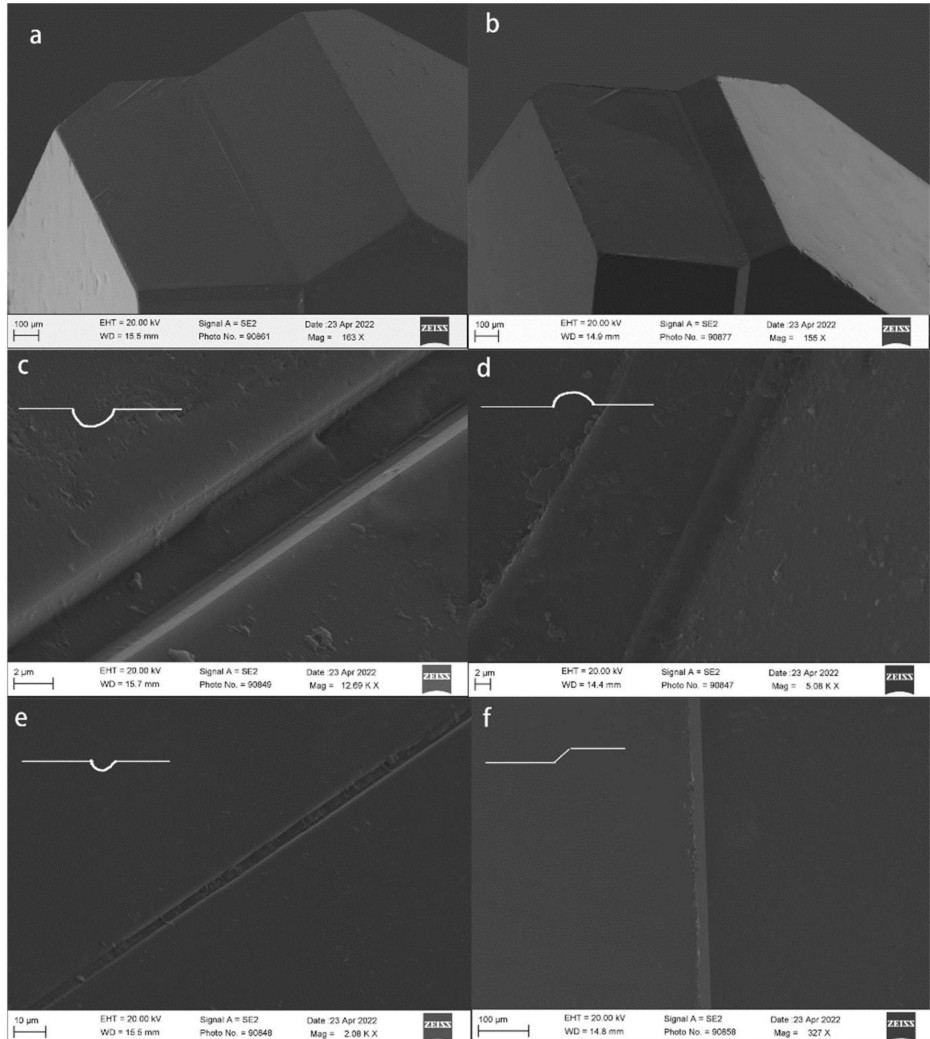

**Figure 4.** The morphology features of the twin boundary observed by SEM, which indicates the twin boundary is not a sharp straight line but a variety of morphological features: (**a**–**e**) the morphology features of the twin boundary are similar to the shape of the pipe with a different height to the crystal plane in TW-01 and TW-07; (**f**) the twin boundary forms a zigzag structure in TW-12.

It can be observed that the twin boundary of the HPHT-grown synthetic diamond crystals with contact twinning was not a simple, straight line extending across the entire crystal. As the outcrops of the twinning plane on the crystal surface, that means the twinning plane has a certain thickness. Guo Qizhi et al. (1992) proposed that the twin boundary was not a perfect plane, but merely a micro-curved surface with different thicknesses between 0–100 μm in different parts [22].

### 3.3. Etch Figures and Related Growth Features

The surfaces microtopography of grown crystals can present a variety of features depending on the different growth conditions. The observation of the surface topography features in detail facilitates the study of the crystal growth mechanism [23–25]. The growth mechanism of HPHT-grown synthetic diamonds includes dislocation (spiral) and non-dislocation (two-dimensional nucleation). V. M. Kvasnytsya (2021) proposed that dislocation (spiral) growth is inherent in synthetic HPHT-grown diamonds, which can reflect the appearance of frequent spirals on cube and octahedron faces. Additionally, the growth mechanism of two-dimensional nucleation is also indicated by frequent skeletal cube and octahedron faces [6]. Tolansky and Sunagawa (1976) recognized that the morphology of synthetic diamond crystals depended on the growth conditions and solvent composition [3,26].

There are three crystallographic directions of diamond lattices, no matter whether they are natural and laboratory-grown diamonds. Therefore, crystal planes can develop different resorption features, which lead to a better understanding of the formation of HPHT-grown synthetic diamonds with contact twinning [27]. In this section, the surface topography features of all the samples were observed by gemological microscopy and 3D Laser Confocal Microscopy, and we selected typical features for discussion to reveal their formation information.

### 3.3.1. Seed Crystals

There are two methods of crystal growth for the HPHT-grown synthetic diamonds. At present, the large-sized diamond crystals are recognized as being grown by the temperature gradient method with seed crystals, whereas the size of the spontaneous nucleation of diamond crystals without seeds is smaller. In the group of HPHT-grown synthetic diamonds with contact twinning, only the TW-01 sample showed seed crystals on the (100) plane and it was corroded (Figure 2g). This feature can reflect the growth process of crystals. During the process of crystal growing, the solvent metal was melted through temperature increments and reacted with the carbon source at the interface to achieve local equilibrium and formed a carbon-containing liquid phase. Before the carbon content in the liquid phase reached the saturation of graphite and supersaturation of the diamond, the solvent metal could etch the seed crystal. Consequently, the etched seed crystal reflected the rate of solvent metal dissolution at an excessive speed [28,29].

### 3.3.2. Etch Figures

Many typical etch figures were observed in the samples. It was observed that the octahedral plane had a nearly flat surface, but the triangular, hexagonal, circular planes and their transition pits were irregularly distributed on it (Figure 5a). They could also cover the whole crystal plane, similar to the fish-scale pattern (Figure 5b). The circular pits did not exist isolated, but were often linked to others (Figure 5c). Some etch pits showed that the point-bottomed different origin compared to the flat-bottomed origin, which, due to the surface defects and foreign particles, caused the of formation for point-bottomed etch pits, presented mixed edges, screw dislocations, or edge dislocations (Figure 5d,e) [30]. Further observations showed that a large number of dense, tiny triangles, which originate from the point defects, were distributed on the {111} plane (Figure 5f). The features of the etch pits revealed that there was a sequence etch process on the octahedral faces. The initial stage of the crystal was etched. Small, triangular etch pits developed, then they became larger and deeper. When the rate of etching increased, the morphology features changed into rounded shapes, then these etch pits could ultimately link to each other [27,30–33].

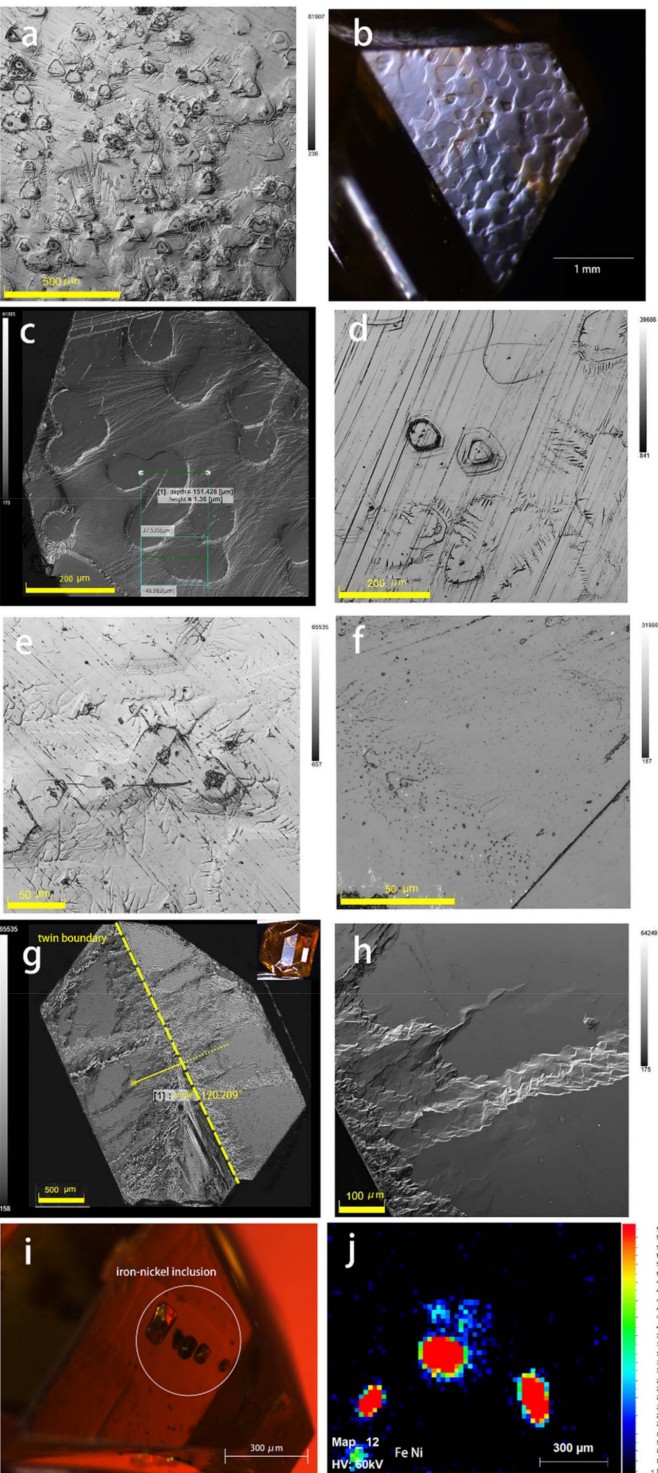

**Figure 5.** Etch figures of the HPHT-grown diamond crystals with contact twinning observed by 3D Laser Confocal Microscopy (**a–h**); (**a**) many kinds of etch pits on crystal plane in TW-07; (**b**) etch pits forms the fish-scale pattern in TW-02; (**c**) circular pits on crystal plane in TW-02; (**d,e**) some etch pits show that point-bottomed origins in TW-02 and TW-07; (**f**) a large number of dense, tiny triangles distributed on the {111} plane in TW-04; (**g,h**) the vein-like pattern on both sides of the twin boundary in TW-01; (**i,j**) the Fe-Ni inclusion in TW-09 and distribution of Fe-Ni elements in micro-XRF testing.

In addition, the surrounding of the etch pits was often distributed in a vein-like pattern. It was observed that the vein-like pattern around the etch pits was fine and had no obvious crystallization direction. It was also observed on both sides of the twin boundary

(Figure 5g,h). The dendritic vein-like pattern was quite common on the crystal plane of the samples presented in this study. The patterns were formed by the precipitation of carbon atoms in the space of the dendrites of the solvent metals during the quenching process, which is one of the characteristic features of synthetic diamonds [34–36]. The research the morphology features of vein-like or dendrite patterns produced an index for the kinds of solvent metals in the growth process. According to the previous studies, the diamond surfaces grown in the Ni or Ni-Fe alloy had dendritic or vein-like patterns [37,38]. As the characteristic patterns correspond to the texture of the quenched solvent metals on the surfaces of the synthetic diamonds, it can be inferred that the solvent metals used in the contact twinning growth were Fe and Ni [37–39]. Moreover, they could also form the metal inclusions within the HPHT synthetically grown diamond crystals (Figure 5i,j).

### 3.3.3. Growth Features

Growth features are rarely observed on HPHT-grown synthetic diamonds due to the quenched solvent metals that form the dendritic pattern covering on the surface of crystal plane. However, when the sample is observed at a high enough magnification, the related growth features on the surface of the crystal plane can be observed, which helps to reveal the growth mechanism of diamond contact twinning. In addition to the features of plentiful etch pits, a special line was observed on the (111) plane, which can extend the entire crystal plane parallel to a crystal edge. A special line was also observed in the synthetic diamond twinning, and sometimes they occurred in combination with a re-entrant angle. N.J. Pipkin named it "ghost line", but did not provide a further explanation [17]. As shown in Figure 6a, it can be observed that the special line was the crystal steps accompanied by a dendritic pattern and the step height was about 12 μm. In some cases, it can penetrate the etch pits (Figure 6b,c), which indicates the etching process occurred before the formation of the crystal steps. In the TW-04 sample, the step height that extends the entire crystal plane was about 0.07 μm and the orientations of the trigons on the both sides of it were consistent (Figure 6b,d), showing that it occurred on the single crystal of contact twinning that did not affect the crystallization of twinning.

By a further magnified observation under SEM, the crystal with the (111) plane exhibited a different character with a lot of jagged, trigonal growth units on the (111) plane (Figure 6e). It was concluded that on the growth surface, the crystal was not grown layer by layer as a whole, but there were a lot of growth sectors growing simultaneously. Additionally, the difference in the unit distance of the jagged trigonal layer can reflect the inhomogeneity of the crystal growth rate [40]. The (100) faces showed the different growth features of (111). The substrate of the cube surface was obviously rougher than that of the octahedron (Figure 6f), which can be explained by the different present states of the surface and inner atoms for (100) and (111) faces. The cube surface had two dangling bonds, while the octahedron surface only had one dangling bond, which means that the interaction between (100) faces atoms and outside atoms or molecules should be stronger than that for (111) surface atoms when, for outside atoms or molecules, the stability of the crystal plane state is weak. Therefore, the ultimate morphology of the crystal plane was rougher than that of the octahedral plane.

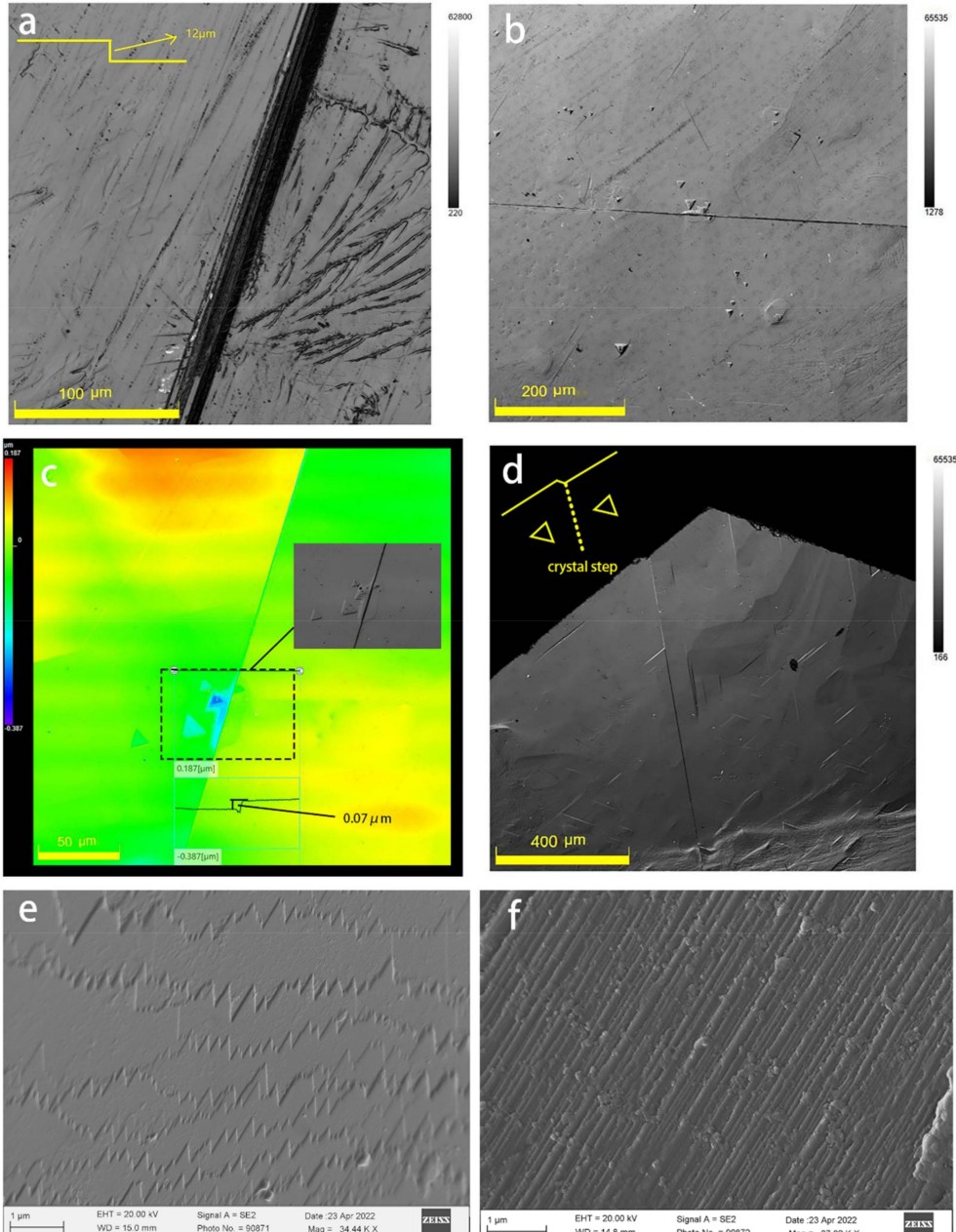

**Figure 6.** Growth features of HPHT synthetically grown diamonds with contact twinning: (**a**) the crystal steps accompanied by a dendritic pattern and the step height is about 12 μm in TW-02; (**b**,**c**) the crystal step can penetrate the etch pits in TW-04; (**d**) the orientations of the trigon pits on both sides of the twin boundary are consistent in TW-10; (**e**,**f**) different growth features for (111) and (100) planes in TW-10 indicate that the substrate of the cube surface is obviously rougher than that of the octahedron.

## 4. Discussion

When a crystal is made up of parts that are oriented with respect to one another, according to some symmetry rules, the crystal is said to be twinned. The common symmetry rule of twinning is reflection rotation and inversion. The formation of twin crystals is a kind of deformation. Diamond twins are only generated during the growth process [5,41]. Compared to the natural diamond, twinning in synthetic diamonds is rare. It may be caused by the following factors. HPHT synthetically grown diamonds with rapid growth rates produce more uncertainty about the location of the carbon atoms' deposition and crystallization. The twinning crystal could form and extend into the inner crystal from the twin nucleus formed in the nucleation process [42], which has a considerable contingency.

From the observation of this set of HPHT synthetically grown diamonds with contact twinning, it can be concluded that there are two types of growth mechanisms that can be distinguished: (i) the temperature gradient method with seed crystals and (ii) spontaneous nucleation. The sample of TW-01 was grown by method (i) due to the fact that it can be observed that the seed crystal on (100) plane and the others are more likely to grow by method (ii). Therefore, TW-01 differs from the other samples in terms of growth conditions, mainly whether seed crystals are used. However, no matter what kinds of growth mechanisms were used, this set of HPHT synthetically grown diamonds with contact twinning all exhibited the features of symmetrical crystals and had re-entrant angles, which is a feature of twin crystals. From the different macroscopic morphologies, the synthetically grown diamond can be distinguished into three types, as shown in Figure 7.

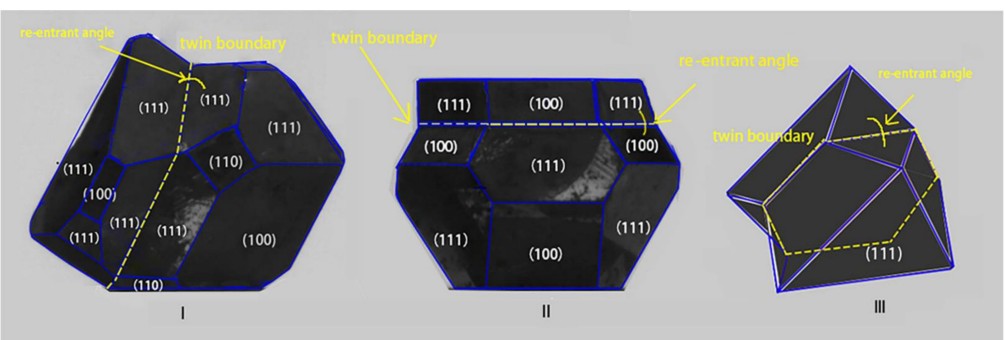

**Figure 7.** The type of contact twinning in HPHT synthetically grown diamond: (**I**) the number and development of {100} and {111} growth sectors on either side of the twin boundary were different in type I; (**II**) the single crystal of the twining contacts the planes of (100) and (111) in type II; (**III**) the crystal morphology seems similar to the Macle in natural diamonds in type III.

Among the 18 samples, the majority of crystal morphologies were type I. The number and development of {100} and {111} growth sectors on either side of the twin boundary were different. Additionally, the crystal domains shared common octahedron faces, were are offset from one another to create surface faceting with re-entrant angles and the twin boundary was a straight and sharp line observed by gemological microscopy, but the morphology features of it changed to the shape of a pipe with a different height from the crystal plane when it was observed by SEM, which means the twin boundary was not a perfect plane but merely a micro-curved surface with different thicknesses. This is consistent with the previous study. Types II and III were less frequent. The single crystal of the twining contacts the planes of (100) and (111) in type II, including TW-07 and TW-12, of which the twin boundary forms a zigzag structure. The crystal morphology of type III, including TW-02 and TW-04, seems similar to the Macle in natural diamonds. The octahedron plane of the synthetic diamond crystals was mainly developed.

In order to visualize the formation process of HPHT synthetically grown diamonds with contact twinning, crystal modeling was built as shown in Figure 8. It can be concluded that types of I and II conform to the model (A). The common crystal morphology of HPHT synthetically grown diamond is cubic-octahedral. When the contact twinning forms, the single crystal rotates 180° along the <111> axis. Additionally, the twin plane is (111). Due to the number and development of the {100} and {111} growth sectors on either side of the twin boundary being different, the symmetry of tower crystals is different. When the octahedron plane of the HPHT synthetically grown diamond crystals were developed, such as type III, they conformed to model (B). The macroscopic morphology feature of contact twinning shows similar features to Macle in natural diamonds.

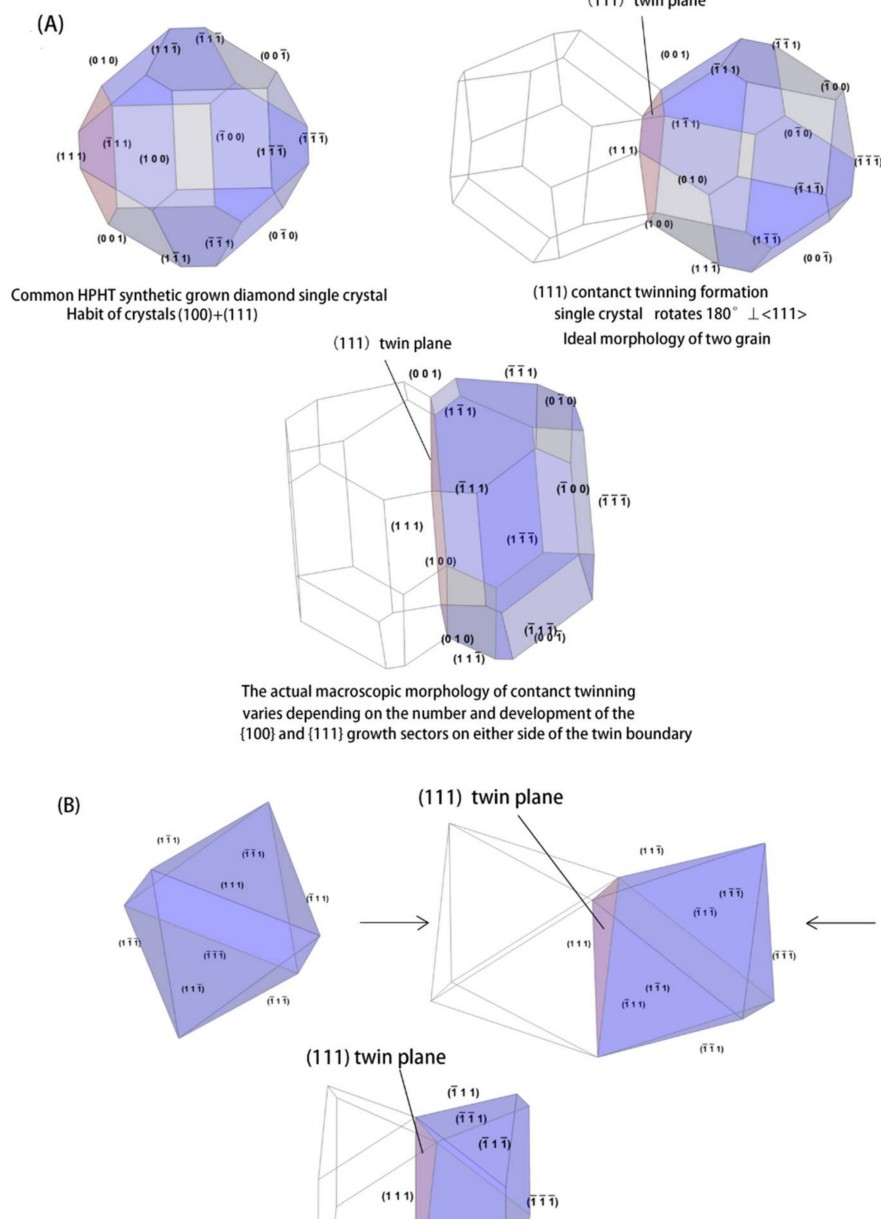

**Figure 8.** Twin models of HPHT synthetically grown diamond with contact twinning: (**A**) The common crystal morphology of HPHT synthetically grown diamond is cubic-octahedral. When the contact twinning forms, the single crystal rotates 180° along the <111> axis and the twin plane is (111); (**B**) When the octahedron plane of the HPHT synthetically grown diamond crystals were developed, The macroscopic morphology feature of contact twinning shows similar features to Macle in natural diamonds.

Combined with the features of etch figures on the crystal plane, there was no obvious crystallography discipline on either side of the twin boundary, which indicates that the development of the {100} and {111} growth sectors on either side of such boundaries seems to proceed independently [17]. The crystallography of twinning is the same, irrespective

of the way twins are produced [41]. The normal arrangement of carbon atom layer in a face-centered cubic (FCC) lattice is ABCABC . . . , but in the contact twinning formation, the carbon atom layer slides along the <121> and the Burgers Vectors is B = a/6 [112] [43]. The arrangement of carbon atom layers is mirror symmetry as shown in Figure 9. From the analysis of the theoretical model of atomic arrangement, the type of lattice at the twin boundary is a hexagonal close-packed (HCP) structure. A previous study determined that the hardness of diamonds can be enhanced by the formation of twin structures, and the ideal indentation strength of a hexagonal diamond surpasses that of a cubic diamond by 58% through the longitudinal sound speeds, which means that the hardness of diamonds with hexagonal close-packed structures increased by more than half [13,44]. Consequently, the contact twinning in HPHT synthetically grown diamonds should have the physical properties of hexagonal diamonds at the position of the twin boundary.

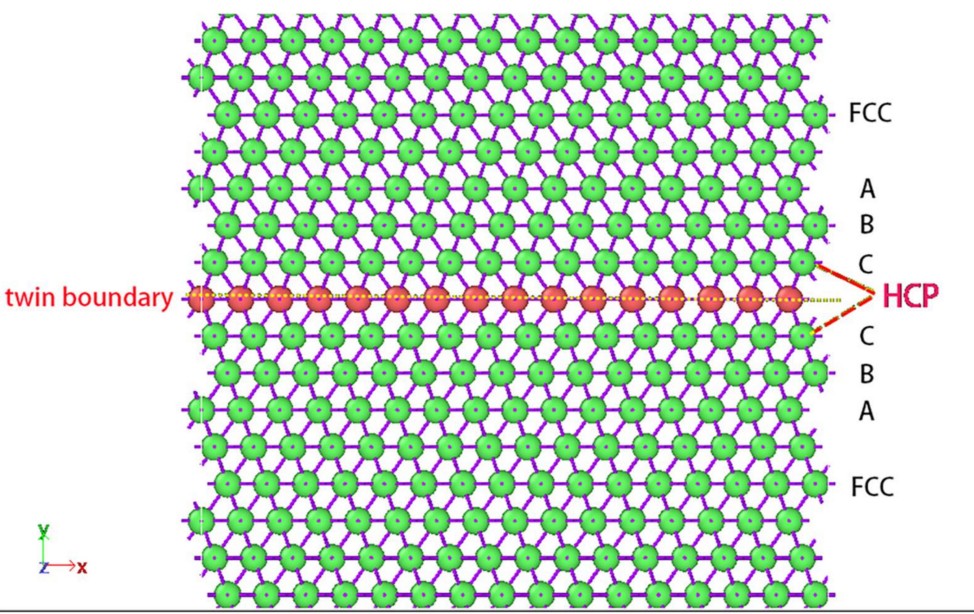

**Figure 9.** The arrangement of carbon atom layers in contact twinning of diamonds is mirrored and the lattice type at the twin boundary is a hexagonal close-packed structure.

## 5. Conclusions

The morphological and surface microtopography features of HPHT synthetically grown diamonds with contact-twinning crystals were studied in this article. Two kinds of twin models of contact twinning in HPHT synthetically grown diamonds were established. From the morphological observation, it can be concluded that twins formed, nucleating during the early stages of the crystal growth process. The single-diamond crystal rotated 180° along the <111> axis. Additionally, the twin plane was (111). The development of the {100} and {111} growth sectors on both sides of the twin boundary proceeded independently, which affected the final morphology of the diamond crystal. The macroscopic morphological feature of contact twinning was similar to Macle in natural diamonds when the octahedron developed.

Due to the fact that HPHT synthetically grown diamonds grow rapidly, this produced further uncertainty about the deposition and crystallization of carbon atoms. Consequently, the formation of twinning was occasional. According to the plentiful surface microtopography features, diamond twinning crystals suffered the strong etching process, which reflected the instability of the growth process. The formation of twin crystals changed the lattice structure of the diamonds. The type of lattice at the twin boundary was HCP, which was beneficial to further develop the application of synthetic diamond twin crystals.

**Author Contributions:** Conceptualization, K.S., T.L. and M.H.; data curation, K.S., T.L. and M.H.; formal analysis, K.S., T.L., M.H., Z.S., J.Z. and J.K.; investigation, K.S.; project administration, T.L. and M.H.; supervision, K.S., T.L., M.H., Z.S., J.Z. and J.K.; writing—original draft, K.S. All authors have read and agreed to the published version of the manuscript.

**Funding:** This research was funded by the National Natural Science Foundation of China (42073008).

**Institutional Review Board Statement:** Not applicable.

**Informed Consent Statement:** Not applicable.

**Data Availability Statement:** The data presented in this study are available within the article.

**Acknowledgments:** Thanks to the National Infrastructure of Mineral Rock and Fossil Resources for Science and Technology and National Gems & Jewelry Testing Co., Ltd. (NGTC). Thanks to the sample providers Fengshun Xu. Thanks to Xuxu Wu, Jinyu Zheng, and Shaokun Wu for their thoughts on the thesis and experiments. We are also grateful to all reviewers and editors for their constructive and helpful comments, which significantly improved the manuscript.

**Conflicts of Interest:** The authors declare no conflict of interest.

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
