# Peer review of "Morphological and Surface Microtopographic Features of HPHT-Grown Diamond Crystals with Contact Twinning"

_crystals, doi:10.3390/cryst12091264_

Round 1

Reviewer 1 Report

Dear Authors

the issue of your paper is very interesting and I’m sure that the studies on the morphologies of diamonds twinned crystals could rise new questions on the complexity of their growth condition especially for HPHT methods.

The samples studied were not labelled so it is difficult to understand the different cases. Moreover, the condition of growth of each sample is not described.

I’m sure that the different morphologies and size of crystals were due to different growth mechanism. Do you have detail about trace elements? Can they contribute to the colour and to the twinning mechanism?

I suggest completing all the figures caption with detailed description of all the pictures that compose the figure. To which samples are related the SEM images?

I suggest to identify the crystals grown by seed from those generated by spontaneous nucleation.

To confirm the presence of two or more individual in twinned crystals the morphologies is not exhaustive in my experience you should have some diffraction data. Can you perform XRD analyses on your samples to confirm the existence of twinning in all yours samples? One crystal looks like an aggregate and in this case, it could be related to a restarting growth.

In the discussion you wrote correctly that the type con contact twinning suggesting two mechanism of growth but were not described different growth conditions for the samples in the paper. Also, the discussion about the “hexagonal diamond” should be proved in this case, not generally, by crystallographic data. 

I suggest to improve your paper adding detailed information about growth conditions and diffraction data.

Regards

Reviewer 2 Report

The present work about twinned crystalline diamond is very interesting and of high quality. I cannot find any need for minor or major corrections, and will therefore recommend this manuscript for publication in the present form.

Reviewer 3 Report

In my opinion, the paper is fairly correctly written, although it unnecessarily puts unscientific information on who is the largest producer of HPHT diamonds, it really doesn't have any scientific significance.

Some comments:

1. The conditions for the formation of the tested samples are poorly described. The HPHT method is well known, but it depends on many factors, not only pressure and temperature. The samples are anonymous, they have been selected, why haven't others been selected, are they samples specific to certain technological processes? Why were these samples selected?

2. The description of test methods should be improved. It is necessary to indicate which diamonds were tested by which methods, it is not enough to write that some.

3. Nowhere have I found an exact indication of which samples the application relates to, it would be optimal to say that a specific sample was produced for certain parameters.

4. In the presented form, the analyzed article is in fact a report on research on a dozen or so diamonds about which we know nothing: how they were made, what properties they have. Maybe it doesn't matter for the subject of the article, but the people who are going to read it are physicists and technologists, they must have facts, figures, descriptions. They will draw their own conclusions as long as they are convinced of your opinions.

Round 2

Reviewer 3 Report

The work looks a lot better, thank you. In the future, please always describe the technological parameters of the tested samples very precisely. This is important for engineers-technologists who work on similar materials.